# TGF-Beta-Activated Cancer-Associated Fibroblasts Limit Cetuximab Efficacy in Preclinical Models of Head and Neck Cancer

**DOI:** 10.3390/cancers12020339

**Published:** 2020-02-03

**Authors:** Ksenia M. Yegodayev, Ofra Novoplansky, Artemiy Golden, Manu Prasad, Liron Levin, Sankar Jagadeeshan, Jonathan Zorea, Orr Dimitstein, Ben-Zion Joshua, Limor Cohen, Ekaterina Khrameeva, Moshe Elkabets

**Affiliations:** 1The Shraga Segal Department of Microbiology, Immunology, and Genetics, 84105 Beer-Sheva, Israelnovoplan@post.bgu.ac.il (O.N.); manupras@post.bgu.ac.il (M.P.); jagadees@post.bgu.ac.il (S.J.); zoreaj@post.bgu.ac.il (J.Z.); ayashli@post.bgu.ac.il (L.C.); 2Faculty of Health Sciences, Ben-Gurion University of the Negev, 84105 Beer-Sheva, Israel; orrdimi@gmail.com (O.D.); BenzionJ@clalit.org.il (B.-Z.J.); 3Center of Life Sciences, Skolkovo Institute of Science and Technology, 121205 Moscow, Russia; golden.artemiy@gmail.com; 4Bioinformatics Core Facility, National Institute for Biotechnology in the Negev, Ben-Gurion University of the Negev, 84105 Beer-Sheva, Israel; levinl@post.bgu.ac.il; 5Department of Otolaryngology-Head & Neck Surgery, Soroka University Medical Center, 84105 Beer-Sheva, Israel

**Keywords:** head and neck cancer, cancer-associated fibroblast, Cetuximab, tumor microenvironment, therapy resistance

## Abstract

Most head and neck cancer (HNC) patients are resistant to cetuximab, an antibody against the epidermal growth factor receptor. Such therapy resistance is known to be mediated, in part, by stromal cells surrounding the tumor cells; however, the mechanisms underlying such a resistance phenotype remain unclear. To identify the mechanisms of cetuximab resistance in an unbiased manner, RNA-sequencing (RNA-seq) of HNC patient-derived xenografts (PDXs) was performed. Comparing the gene expression of HNC-PDXs before and after treatment with cetuximab indicated that the transforming growth factor-beta (TGF-beta) signaling pathway was upregulated in the stromal cells of PDXs that progressed on cetuximab treatment (Cetuximab^Prog^-PDX). However, in PDXs that were extremely sensitive to cetuximab (Cetuximab^Sen^-PDX), the TGF-beta pathway was downregulated in the stromal compartment. Histopathological analysis of PDXs showed that TGF-beta-activation was detected in cancer-associated fibroblasts (CAFs) of Cetuximab^Prog^-PDX. These TGF-beta-activated CAFs were sufficient to limit cetuximab efficacy in vitro and in vivo. Moreover, blocking the TGF-beta pathway using the SMAD3 inhibitor, SIS3, enhanced cetuximab efficacy and prevented the progression of Cetuximab^Prog^-PDX. Altogether, our findings indicate that TGF-beta-activated CAFs play a role in limiting cetuximab efficacy in HNC.

## 1. Introduction 

Over half a million new cases are diagnosed each year with head and neck cancer (HNC), making HNC the seventh most common form of malignancy worldwide [1]. A combination of traditional regimens of radiotherapy and chemotherapy, with Cetuximab (Erbitux), is a common clinical practice in the care of patients with advanced or metastatic HNC [2,3,4,5]. Cetuximab is a monoclonal antibody designed to block ligand binding to the epidermal growth factor receptor (EGFR) ([6] and reviewed in [7]). EGFR is a transmembrane protein that, upon ligand binding, stimulates its intracellular tyrosine kinase activity that subsequently activates the mitogen-activated protein kinase (MAPK) pathway [8]. In HNC, EGFR is a key receptor that is known to be involved in tumor progression and metastasis and its expression associated with poor prognosis [9,10,11]. Despite the pivotal role of EGFR in HNC, blocking EGFR using cetuximab showed an overall minimal effect, and only a small fraction of HNC patients benefit from such treatment. Moreover, the majority of patients who initially responded will develop resistance to cetuximab and experience disease relapse [5,12].

Currently, the resistance mechanisms to cetuximab are known to be mediated by both intracellular signaling machinery in tumor cells (cell-autonomous) and secreted factors by cells in the tumor microenvironment (TME) (nontumor-cell autonomous) (reviewed [13]). The tumor cell-autonomous mechanisms include mutations in the EGFR or downstream-related genes [14,15,16], as well as activation of compensatory loops such as human epidermal growth factor receptors 2 and 3 or AXL (HER2 and HER3, respectively) [17,18,19,20]. The nonautonomous resistance mechanisms include secreted factors from cellular components in the TME, such as cancer-associated fibroblasts (CAFs). In vitro, CAFs were shown to confer resistance to cetuximab in HNC cell lines via matrix metalloproteinase-1 (MMP-1) and hepatocyte growth factor (HGF) [21,22]. In biopsies of HNC patients, the response to cetuximab was shown to be associated with cetuximab-induced CAFs activation [23]. Moreover, the immunosuppressive environment in the tumor can reduce cetuximab-induced antibody-dependent cell-mediated cytotoxicity (ADCC), as secreted transforming growth factor-beta (TGF-beta) can reduce natural killer (NK) cells’ activity [24,25]. Although the TME’s role in therapy resistance has been observed in cancer patients and blocking components in the TME improves therapy efficacy [26,27,28,29], to date, the mechanisms underlying CAFs activations and their role in conferring resistance to cetuximab in vivo have not yet been reported.

In this work, we sought to investigate the crosstalk between tumor cells and the TME following cetuximab treatment. By comparing the response of patient-derived xenografts (PDXs) to cetuximab in vivo, we demonstrated that progression to cetuximab is associated with increased TGF-beta signaling by CAFs. TGF-beta-activated CAFs induced cetuximab resistance in vitro and in vivo. Moreover, blocking the TGF-beta signaling was sufficient for enhancing the efficacy of cetuximab in PDX-bearing mice.

## 2. Results

### 2.1. The Diverse Response of HNCs-PDXs to Cetuximab Treatment

Five PDXs (PDXs information, Appendix A) were implanted in NOD/SCID mice to determine the antitumor effect of cetuximab on PDX progression in vivo. When tumors reached ~100 mm^3^, the mice were randomized into two arms treated either with vehicle or cetuximab. The percentage of change in tumor volume following cetuximab treatment (Figure 1A) and the tumor growth kinetics (Appendix A) indicate that all five PDXs responded to cetuximab. However, in three of the PDXs, namely, PDX #01, PDX #03, and PDX #20, cetuximab induced tumor shrinkage, resulting in decreased tumor volume (this will be referred to hereafter as Cetuximab^Sen^-PDXs, Figure 1A, right). In contrast to the Cetuximab^Sen^-PDXs, in the other two PDXs, namely, PDX #18 and PDX #19, a progressive disease was observed, resulting in increased tumor volume compared with the starting point (this will be referred to as Cetuximab^Prog^-PDXs, Figure 1A, left). Pathological analyses of all PDXs by immunohistochemistry (IHC) were performed to quantify the on-target effect of cetuximab by staining against phosphorylated MAPK (pMAPK), a downstream protein of EGFR. In addition, the proliferation rate was assessed by staining with Ki67. A reduction in pMAPK and Ki67 levels following cetuximab treatment was observed in all PDXs (Figure 1B,C and Appendix A). However, such analysis was inconclusive to distinguish between Cetuximab^Sen^-PDXs and Cetuximab^Prog^-PDXs. 

### 2.2. Molecular Characterization of Cetuximab^Sen^ and Cetuximab^Prog^ PDXs

To gain further molecular insight into the mechanisms underlying the response to cetuximab, a bulk RNA-sequencing (RNA-seq) of PDXs treated with cetuximab or vehicle was performed. Specifically, for the sequencing, two PDXs that exhibited tumor shrinkage, PDX #03 and PDX #20, and a single PDX, PDX#18, which exhibited disease progression, were selected (Figure 2A). The obtained sequencing reads, which uniquely mapped to a concatenated human and mouse genome, were separated into mouse reads and human reads (see Methods). Multidimensional scaling (MDS) analysis of mouse and human reads was performed to characterize all three PDXs. The MDS plots show a clear separation of all three PDXs based on the human reads (Figure 2B, left), but, in the murine reads, the distance between cetuximab^Sen^-PDX tumors was lower compared with Cetuxiamb^Prog^-PDX (Figure 2B, right). Moreover, upon treatment with cetuximab, the expression of mouse genes (stromal compartment) was changed in Cetuximab^Sen^-PDXs but to a lesser extent in Cetuxiamb^Prog^-PDX (Figure 2B, right). A direct comparison of treatment-induced gene expression changes between PDXs (Appendix A) and differential expression analysis (Appendix A) revealed a similar effect.

To investigate the commonly enriched pathways that were upregulated in Cetuxiamb^Prog^-PDX and downregulated in Cetuximab^Sen^-PDXs, and vice versa, pathway enrichment analysis of the stromal compartment, based on the KEGG annotation database [30], was performed. Forty-four pathways enriched with genes significantly upregulated in the PDX #18, and 178 and 168 pathways were downregulated in PDX #20 and PDX #03, respectively (the negative binomial test, |log2FC| > 0.5, BH-corrected p-value < 0.05, Figure 2C and Appendix A). Interestingly, there were 155 pathways in common between the two Cetuximab^Sen^-PDXs, whereas there were only zero and five pathways in the Cetuximab^Sen^-PDXs/Cetuxiamb^Prog^-PDX comparisons (Figure 2C). Eight pathways (Table 1) exhibited enrichment with genes upregulated in PDX #18 and downregulated in both PDX #20 and #03 (complete table of signatures, Appendix A). The TGF-beta pathway was among those eight pathways, showing upregulation in Cetuxiamb^Prog^-PDX and downregulation in Cetuximab^Sen^-PDXs (Figure 2C). As the TGF-beta signaling pathway was so prominent in the enriched signatures of Cetuxiamb^Prog^-PDX and TGF-beta was shown to limit cetuximab efficacy [31], this pathway was chosen for further investigation as a potential candidate involved in the progression phenotype under cetuximab treatment. 

### 2.3. Tumor Progression Under Cetuximab Treatment is Associated with TGF-Beta Activation in Cancer-Associated Fibroblasts (CAFs)

The RNA-seq results regarding the changes in the TGF-beta pathway within the stroma of Cetuximab^Sen^-PDXs and Cetuximab^Prog^-PDX were validated using IHC staining for the phosphorylated SMAD family member 2 (pSMAD2). Nuclei staining of pSMAD2 indicates the activation of the TGF-beta signaling pathway [32]. Quantification of the pSMAD2 expression level in the stroma revealed a reduction following cetuximab treatment in Cetuximab^Sen^-PDXs (Figure 3A, right), indicating the downregulation of the TGF-beta signaling pathway in the TME. In contrast to Cetuximab^Sen^-PDXs, elevated expression levels of pSMAD2 were detected in the stroma of Cetuximab^Prog^-PDXs after cetuximab treatment (Figure 3A, left and Appendix A), indicating upregulation of the TGF-beta pathway. An example of stromal analysis is shown in Appendix A. As pSMAD2 was expressed in spindle-shaped stromal cells corresponding to the shape of cancer-associated fibroblasts (CAFs) [33], the CAF’s marker, alpha-smooth muscle actin (α-SMA) [34], and pSMAD2 were costained in tumor sections. The coexpression of pSMAD2 in the nuclei of α-SMA-expressing cells confirmed that the TGF-beta pathway is activated in CAFs and tumor cells (Figure 3B). 

To explore the direct role of TGF-beta-activated CAFs in limiting the efficacy of cetuximab, culturing of CAFs from Cetuximab^Prog^-PDX (PDX #19) was established. These CAFs were positive for pSMAD2 in the nuclei, confirming the TGF-beta pathway activation, compared to the immortalized fibroblasts of the NIH 3T3 cell line and primary cultured murine normal fibroblasts (NOFs) (Appendix A). To explore if secreted factors from TGF-beta-activated CAFs can limit the efficacy of cetuximab in vitro, tumor cell proliferation was measured in the presence of cetuximab with or without conditioned media (CM) collected from CAFs. Specifically, the proliferation of the cetuximab-sensitive HNC cell line, Detroit562, showed that the CM of the TGF-beta-activated CAFs increased tumor cell proliferation (Appendix A) and decreased their sensitivity to cetuximab, compared with the response of the cells with normal media (Figure 3C).

### 2.4. TGF-Beta-Activated CAFs Reduce Sensitivity to Cetuximab In Vivo and Blocking TGF-Beta Sensitize HNC-PDX to Cetuximab 

To study the effect of TGF-beta-activated CAFs on HNC cell line growth in vivo, the cetuximab-sensitive cell line CAL33 [35,36] was injected subcutaneously into NOD/SCID mice, with or without TGF-beta-activated CAFs. CAFs promoted tumor growth in a ratio-dependent manner, compared with CAL33 alone, indicated by tumor volume (Appendix A). To determine whether TGF-beta-activated CAFs can limit cetuximab efficacy in vivo, a four-arm experiment was designed. Two groups of mice were injected with only CAL33 tumor cells, and two groups of mice were injected with a mix of tumor cells and CAFs in a ratio of 1:1 (low ratio was used to minimize the effect of CAFs on tumor growth). When tumors reached ~100 mm^3^, treatment with cetuximab or vehicle was initiated. The kinetics of tumor growth (Figure 4A) shows that TGF-beta-activated CAFs limited the efficacy of cetuximab treatment, since cetuximab was less potent in mice with a mix of CAFs and CAL33, compared with tumors in mice with only CAL33 (Figure 4A and Appendix A). These results were also reflected by analyzing tumor weight and comparing the H&E staining of the different groups (Figure 4B,C). These in vivo results further support the trend that was observed in vitro, in which CM from CAFs was sufficient to reduce sensitivity to cetuximab treatment. 

To determine whether inhibition of the TGF-beta signaling pathway will sensitize tumors to cetuximab treatment, the growth of Cetuximab^Prog^-PDX (PDX #18) was measured after treatment with cetuximab, SIS3 (an inhibitor of the TGF-beta pathway) [37], or a combination of both. SIS3 is a specific inhibitor of SMAD3 that was shown to inhibit myofibroblast differentiation of fibroblast by TGF-beta-1 [37]. SIS3 alone delayed the progression of the tumor; however, an increase in tumor volume was observed with time. A similar tumor progression was detected with cetuximab treatment alone. In contrast, coblocking the TGF-beta pathway using SIS3, together with cetuximab, succeeded in delaying the tumor progression, and no change in Cetuximab^Prog^-PDX volume was observed with time (Figure 4D). Tumor weight was lower in mice treated with the combination, compared with the single agents (Figure 4E,F). 

## 3. Discussion

The current treatment options for HNC patients with advanced local or metastatic disease have severe consequences on the quality of life as well as limited effects on these patients’ overall survival [38,39]. Cetuximab was approved for HNC treatment, combined with radiation or chemotherapy, based on the improvement in response rates and a slight improvement in overall survival [2,3,4,5]. Unfortunately, the vast majority of patients experience recurrent or progressive disease [4,12]. The resistance to cetuximab can come in different forms, either by cell-autonomous resistance mechanisms (e.g., gene mutations, receptors upregulation, and feedback loops) or by the cells in the TME, which provide an extrinsic resistance mechanism (for example, secretion of growth factors) ([21,35,40,41] and reviewed in [42]).

In our study, we used an unbiased approach of RNA sequencing to identify the mechanisms of response and resistance to cetuximab. Gene expression profiling of PDXs distinguished between Cetuximab^Sen^-PDXs and Cetuximab^Prog^-PDX by the changes of stromal genes following cetuximab treatment (Figure 1 and Figure 2). A limitation of our approach is the low number of samples. However, the depth of the RNA-seq allowed the identification of significant candidates that were validated in vitro and in vivo. Specifically, the TGF-beta signaling pathway was upregulated after cetuximab treatment in the stroma of Cetuximab^Prog^-PDX, whereas in the two Cetuximab^Sen^-PDXs, downregulation of the same pathway was observed (Figure 2C). TGF-beta plays a crucial role in TME, mediating both immunosuppression and inducing epithelial-mesenchymal transition (EMT) [43,44], and is known to be secreted by CAFs (Reviewed in [45,46]). CAFs are an important component in the TME that play a pivotal role in tumor cell proliferation, invasion, and metastasis in multiple cancer types [47,48,49,50] including HNC [51,52]. CAFs are also known to confer resistance to anti-EGFR therapies among different cancer types including colon [53], lung [54,55,56], breast [57], and in HNC. In HNC cell lines, only one report indicated that CAF-mediated MMPs and secrete -HGF limit the efficacy to cetuximab in vitro [21]. In a different work, involving gene expression analysis of biopsies before and after treatment with cetuximab, an EMT signature was upregulated in tumor cells and an enriched signature of CAF genes was detected in HNC patients that did not respond to cetuximab [23]. These evidences prompted us to further explore the role of CAFs in cetuximab resistance in our PDXs pre-clinical models. Using CAFs isolated from Cetuximab^Prog^-PDX we were able to demonstrate that in addition to enhancing HNC cell proliferation, CAFs reduce sensitivity to cetuximab both in vitro and in vivo (Figure 3C and Figure 4A). We speculate that the mechanism by which CAFs interfere with cetuximab treatment involves the secretion of HGF, which was previously reported to be a mechanism of resistance [21,35,58]. However, the role of TGF-beta-activated CAFs in HGF secretion as a mechanism of resistance to cetuximab calls for further investigation. 

Besides the involvement of CAFs in promoting tumor cell proliferation, and thus limiting the efficacy of cetuximab, CAFs were shown to reduce NK mediating ADCC by downregulation of the NKG2D receptor [59]. A similar reduction in NKG2D expression and a reduction in ADCC can be induced by TGF-beta [31,60,61,62], which can also be secreted by CAFs (reviewed in [44,63]). Our results suggest a similar link between TGF-beta and NK cell activity, as in addition to the upregulation of the TGF-beta signaling pathway in Cetuximab^Prog^-PDX, a downregulation of NK cell-mediated cytotoxicity pathway was found in the RNA-seq. Moreover, upregulation of the NK cell-mediated cytotoxicity pathway was observed in Cetuximab^Sen^-PDXs following cetuximab treatment. Further investigation is required to confirm the effect of TGF-beta activated CAFs on NK cell activity in response to cetuximab treatment. 

TGF-beta has been an attractive target in medical oncology (reviewed in [64]). Here, we demonstrated that blocking TGF-beta signaling by SIS3 can enhance cetuximab efficacy in HNC. A similar therapeutic combination of cetuximab with anti-TGF-beta antibody was previously tested [31]. Specifically, stimulation of tumor cells with recombinant TGF-beta limited the efficacy of anti-EGFR agents erlotinib and cetuximab in vitro, and blocking TGF-beta in vivo consequently enhances sensitivity to cetuximab in HNC model [31]. 

Altogether, our work shows that resistance to cetuximab is associated with increased TGF-beta pathway activation in CAFs. These TGF-beta activated CAFs secrete factors that limit the antitumor activity of cetuximab in vitro and in vivo. Blocking TGF-beta signaling was sufficient to enhance cetuximab efficacy and prevent HNC progression (Scheme 1).

## 4. Materials and Methods

### 4.1. Sample Procurement

Fresh tumor tissue samples from five HNC patients were procured after surgery. The samples were placed in phosphate-buffered saline 1x (PBS) for transport and then processed within 2–3 h after harvesting. Samples were collected with the patient’s consent and with Helsinki approval from the Ear Nose and Throat Unit, Soroka Medical Center, Israel (ethic code 0421-16-SOR and 0103-17-SOR).

### 4.2. Mice and Establishment of Patient-Derived Xenografts (PDXs)

All experiments were conducted using 6–8-week-old NOD/SCID mice (Envigo, Huntingdon, UK, NOD.CB17-Prkdcscid/NCrHsd). Forming the PDX model, patient-derived tumor tissue samples were implanted subcutaneously in the dorsal flanks of the mice. The tumor growth rates varied from 1 to 6 months. PDXs have been maintained by passing the tumors in mice from the first generation to subsequent generations. Upon successful tumor engraftment, the tumors were expanded and retransplanted into NOD/SCID mice for the drug efficacy experiments.

### 4.3. In-Vivo Experiments

Five PDXs were transplanted in NOD/SCID mice. For each PDX, 3 mm^3^ pieces of tumors were implanted. Each group contained 3–6 mice harboring two tumors (*n* = 6–12). When tumor volume reached 70 to 120 mm^3^, the animals were randomly divided into two arms treated either with vehicle or cetuximab. In the cell line-derived xenograft (CDX) experiment, CAL33 cells with or without CAFs were injected subcutaneously into NOD/SCID mice. In all experiments, tumor measurement was performed with a digital caliper, and tumor volumes were determined with the formula: length × width^2^ × (π/6). All animal experiments were carried out under the Institutional Animal Care and Use Committee (IACUC) of Ben-Gurion University of the Negev (BGU’s IACUC) according to specified protocols aimed to ensure animal welfare and to reduce suffering. The animal ethical clearance protocol numbers used for this study are IL-80-12-2015 and IL-29-05-2018.

### 4.4. Chemical Compounds

The EGFR inhibitor, Cetuximab (Erbitux, Merck, Kenilworth, NJ, USA), was kindly provided by Soroka Medical Center. The drug was diluted with sterile PBS (1x) for a working concentration of 10 mg/kg for in vivo and 12.5 μg/mL for in vitro experiments. Cetuximab was given in vivo by intraperitoneal injection (i.p.) every five days. The SMAD3 inhibitor, SIS3 (APExBIO, Houston, TX, USA), was dissolved in 0.5% DMSO and then with corn oil for a final working concentration of 5 mg/kg. SIS3 was given in vivo by IP every day. Recombinant mouse TGF-beta1 (BioLegend, San Diego, CA, USA) was used for the in vitro experiments in a 10 ng/mL working concentration. 

### 4.5. Library Preparation and RNA Sequencing (RNA-seq)

Sequencing was performed on three out of five PDXs (PDX #03, #18, and #20). Three tumors from each group (vehicle and cetuximab, a total of 6 samples for each PDX) were subjected to sequencing. Tumors were obtained from different mice within the group. RNA was extracted using the RNeasy mini kit (Qiagen, Hilden, Germany) according to the manufacturer’s instructions. After quality control of Bioanalyzer (RIN > 7), RNA-seq libraries were prepared using the TruSeq RNA Sample Preparation kit (Illumina, San Diego, CA, USA) according to the manufacturer’s protocol. Briefly, 1ug of total RNA was fragmented, followed by reverse transcription and second-strand cDNA synthesis. The DS-cDNA was subjected to end repair, a base addition, adapter ligation, and PCR amplification to create libraries. Qubit and Bioanalyzer were evaluated. Sequencing libraries were constructed with barcodes to allow multiplexing of 6 samples in one lane. Libraries, ready to run, were sent to the g-INCPM institute at the Weizmann Institute of Science for sequencing. On average, 400 million paired-end (75 bp) reads were sequenced per run on an Illumina NextSeq 500 HO instrument.

### 4.6. RNA-Seq Data Analysis

Sequencing reads were mapped to a concatenation of human and mouse reference genomes, and then uniquely mapped reads were separated into human and mouse reads using a pipeline NeatSeq-Flow [65]. The obtained read counts per gene were further processed using the edgeR package [65]. Low-covered genes were filtered out, read counts were normalized, and differentially expressed genes were found using the default parameters. After filtration, we obtained, on average, 39.7 million human reads and 11.2 million mouse reads per sample. As was demonstrated previously [66], coverage of 1.3–2.0 million reads was sufficient to accurately measure gene expression. In this study, high coverage of 1.5–56.6 million reads was obtained to ensure the robustness of the generated data set. Multidimensional (MDS) plots were constructed based on 500 genes with the highest gene expression variation in human or murine. For comparison of treatment-induced gene expression changes between patients, only genes with |log2FC| > 0.5 were taken into account. Log2FC plots were prepared by plotting the fold-change of gene expression for each pair of PDXs against each other on different axes, murine, and human compartments, separately. Genes were filtered by fold-change before plotting, and genes that were differentially expressed with BH-corrected p-values < 0.05 were highlighted in red. Spearman correlations were calculated for each pair of PDXs. KEGG pathway enrichment analysis was carried out with the clusterProfiler R package [67]. RNA-seq data is available in the GEO database (accession number GSE143279). 

### 4.7. Staining: Immunohistochemistry (IHC), Opal, Immunofluorescence (IF), and Hematoxylin & Eosin Staining (H&E)

For IHC: After mice were euthanized, dissected tumors were fixed in 4% paraformaldehyde (PFA) solution overnight at room temperature. The tumors were transferred into cassettes and maintained in 70% ethanol until they were embedded in paraffin. Paraffin-embedded tumor blocks were sectioned at 5 μm, loaded onto microscope slides, and deparaffinized at 65° for 1 h. After additional deparaffinization using a Xylene substitute (Leica Biosystems Inc, 3803672E, Buffalo Grove, IL, USA) and rehydration steps using a descending alcohol solution, antigen retrieval was performed. The slides were incubated in 10mM citric acid buffer, pH 6.0 at 100 °C for 15 min, then cooled in buffer at room temperature and rinsed with double-distilled water (DDW). Slides were kept in 0.3% hydrogen peroxide (H_2_O_2_) in methanol for 30 min for inactivation of endogenous peroxidases. Slides were washed with PBS and then incubated with a blocking solution, 5% BSA in PBS-T (0.1% TWEEN) for 1 h at room temperature. Primary antibodies: Ki67 (Cell marque corporation, Rocklin, CA, USA, 275R-14, 1:200); pMAPK (CST, Denver, MA, USA, 4370S, 1:100); and pSMAD2 (MilliporeSigma, Burlington, MA, USA, AB3849-I, 1:2000) were diluted in blocking solution according to the manufacturer’s recommended concentrations. Slides were incubated overnight at 4 °C with the primary antibodies. The next day, slides were washed with PBS-T, and the ABC kit (Vector laboratories, Burlingame, CA, USA, VECTASTAIN®ABC, VE-PK-6200) was used for detection according to the manufacturer’s protocol, using DAB (ScyTek laboratories, Logan, US, ACH500-IFU) as a substrate for color development. After DAB, slides were costained with hematoxylin (Leica Biosystems Inc, Buffalo Grove, IL, USA), dehydrated, and mounted with mounting media (Leica Biosystems Inc, Buffalo Grove, IL, USA, Sub-X, 3801740).

For Opal: Multiplexing staining protocol using opal reagents is based on simple IHC staining, as detailed above, with a different detection method in a few cycles. α-SMA (Abcam, Cambridge, UK, ab5694, 1:100) and pSMAD2 (MilliporeSigma, Burlington, MA, USA, AB3849-I, 1:500) were used as primary antibodies. After washing out the primary antibody, rabbit horseradish peroxidase (HRP)-conjugated secondary antibody (Jackson ImmunoResearch, PA, USA, 1:500) was used, followed by detection with opal reagents (PerkinElmer, Waltham, MA, USA, Opal™ 520; FP1487001KT, Opal™ 690; and FP1497001KT) according to the manufacturer’s protocol. Slides were mounted with DAPI Fluoromount-G^®^ (SouthernBiotech, Birmingham, MA, USA, 0100-20).

For IF: Fibroblast cell lines (3T3 NIH, CAFs, and NOFs) were seeded on top of cover glass (BarNaor, Petah-Tikva, Israel) in 24-well plates for 24 h with either control or TGF-beta-1 treatment. The media was discarded, and cells were washed with PBS-T and fixed in 4% PFA for 30 min at room temperature. After additional washing with PBS-T, cells were permeabilized for 10 min at room temperature with 0.1% Triton X-100 (MilliporeSigma, Burlington, MA, USA) in PBS-T. Thereafter, cells were washed with PBS-T and blocked in blocking solution (5% BSA in PBS-T (0.1% TWEEN)) for 1 h at room temperature. Next, the cells were incubated with the primary antibody pSMAD2 (MilliporeSigma, Burlington, MA, USA, AB3849-I, 1:500) overnight at 4 °C. The next day, cells were rinsed with PBS-T and incubated with Alexa Fluor-647 Anti-Rabbit secondary antibody (Jackson ImmunoResearch, PA, USA, 111-605-144, 1:250) at room temperature for 1 h. Finally, cells were washed with PBS-T and mounted with DAPI Fluoromount-G^®^ (SouthernBiotech, Birmingham, CA, USA, 0100-20).

For H&E: After deparaffinization using Xylene substitute and rehydration steps, as was done in IHC and opal staining, slides were stained with hematoxylin (Leica Biosystems Inc, Buffalo Grove, IL, USA) for 3 min, followed by extensive rinsing in water. Slides were then subjected to costaining with eosin (Leica Biosystems Inc, Buffalo Grove, IL, USA) for 2 min. Finally, slides were immersed in ascending alcohol solutions, xylene, and then mounted. All slides were digitalized using the Panoramic Scanner (3DHISTECH, Budapest, Hungary). Slides were analyzed by QuantCenter (3DHISTECH) software.

### 4.8. Cell Lines

The HNC cell line Detroit562 (ATCC), CAL33 (DSMZ), and the fibroblast cell lines 3T3 NIH (ATCC), CAFs, and NOFs (generated in the lab) were maintained at 37 °C in a humidified atmosphere of 5% CO_2_. CAFs were isolated from Cetuximab^Prog^-PDX (PDX #19), and NOFs were isolated from wild-type (WT) C57 BL6 mouse lips by fine chopping of the tissue and incubation with trypsin for 10 min at 37 °C. Warm, full 10% media (DMEM) was added to the chopped tissue. The mixture was well pipetted and centrifuged. Pellet was seeded. Cell media was replaced after 24–48 h, and primary cultured cells were grown until plates (60mm) became 80% confluent. Using differential trypsinization, the first cells detached (after 2–3 min) from the plate were seeded separately; these cells represented the CAFs population. Conditioned media from these cells were produced by seeding the cells in a 100 mm plate with 10% media. After 24 h, the plates were washed twice with PBS, and the media was replaced by 1% media. CM was collected and passed through a 70 µm strainer after 24 h. 

### 4.9. Cell Proliferation Assay

Forty-thousand cells were seeded in 24-well plates and treated for four days with Cetuximab (12.5 µg/mL) in the presence or absence of CAFs’ CM. Cells were stained with Crystal Violet (1 g/L). Quantification was carried out by dissolving the crystal violet with 10% acetic acid and by reading at OD 570 nm in a spectrophotometer (Epoch, BioTek, Winooski, VT, USA). 

### 4.10. Statistical Analysis

Statistical analysis was performed using GraphPad Prism software (version 7), presented as the mean ± SEM. For comparisons between two groups, *p* values were calculated by unpaired *t*-test. *p* values of 0.05 (*), 0.01 (**), 0.001 (***), and 0.0001 (****) were considered statistically significant.

## 5. Conclusions

Our work shows that resistance to cetuximab is associated with increased TGF-beta pathway activation in CAFs. These TGF-beta-activated CAFs secrete factors that limit the antitumor activity of cetuximab in vitro and in vivo. Blocking TGF-beta signaling was sufficient to enhance cetuximab efficacy and prevent HNC progression (Scheme 1).

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
