# Peer review of "TGF-Beta-Activated Cancer-Associated Fibroblasts Limit Cetuximab Efficacy in Preclinical Models of Head and Neck Cancer"

_cancers, 2020, doi:10.3390/cancers12020339_

Round 1

Reviewer 1 Report

Yegodayev et al studied the effect of TGF-beta-activated CAFs on the cetuximab activity in head and neck cancer. The topic of research is important as the resistant to cetuximab is a major problem in head and neck cancer treatment. The assays are applied correctly and the results are presented and discussed clearly. There are some issues that need to be addressed before accepting this manuscript:

1) For the RNA sequencing assay, why only one PDX (18) was used for the disease progression group? Unfortunately this makes the results very week as it is very possible that the difference might came by chance and not real. There should be at least another PDX in the assay to have two for each group.

2) In Figure 4, the authors claimed that adding CAF led to reduce in the effect of cetuximab “The kinetics of 203 tumor growth (Figure 4A) shows that TGF-beta-activated CAFs limited the efficacy of cetuximab 204 treatment, since cetuximab was less potent in mice with a mix of CAFs and CAL33, compared with 205 tumors in mice with only CAL33.”

In fact this is not clear from the Figure because CAF was able to increase the size of the tumour so it makes sense that the size of the tumour after adding cetuximab is still larger than the group which have cetuximab without CAF. If the claim of the authors is correct, then the authors should measure the percentage of the reduction in the tumour size after adding cetuximab compared with the vehicle group. The same percentage should be calculated in the other group (Call33+CAF-cet VS Call33+CAF-vehcile). Then a comparison between the percentages of the reduction between these two groups should be done.

3) In the method section, the dilution or the working concentration of the antibodies should be mentioned to allow repetition of the work.

4) There are some grammar mistakes and typos that should be corrected, e.g. “However, such an analysis”, “Forth-four”, and “low ration was”.

Author Response

Also attached at the link below (word file).

--------------------------------------------------------

Dear Editors and Reviewers;

Re: Revised version of the manuscript cancers-702471

Editor’s comments:

We gratefully thank you for inviting us to submit a revised version of the manuscript cancers-702471 addressing the reviewers’ concerns. We have analyzed the Reviewers’ comments carefully. Our response to the comments is given below.

Response to reviewers’ comments, point by point:

We are grateful to the two reviewers for their encouraging and thoughtful comments and suggestions regarding our original submission. In response to these comments, we have made several modifications to our article as suggested. Below we detail the modifications made in the revised version in response to the specific comments of the reviewers, point by point, in dark blue. We hope the reviewers will find the manuscript improved following these changes and suitable for publication in Cancers.

Reviewer 1

Yegodayev et al studied the effect of TGF-beta-activated CAFs on the cetuximab activity in head and neck cancer. The topic of research is important as the resistant to cetuximab is a major problem in head and neck cancer treatment. The assays are applied correctly, and the results are presented and discussed clearly. There are some issues that need to be addressed before accepting this manuscript.

Answer: We sincerely appreciate the reviewer's comments. Cetuximab resistance is a major challenge in the treatment of HNC patients. We invested a lot of work and effort in order to shed light on the underlying mechanism of cetuximab resistance.

For the RNA sequencing assay, why only one PDX (18) was used for the disease progression group? Unfortunately, this makes the results very week as it is very possible that the difference might came by chance and not real. There should be at least another PDX in the assay to have two for each group.

Answer: We thank the reviewer for raising this valuable point, as sequencing another CetuximabProg-PDX would be informative to further statistical analysis. Unfortunately, due to financial constraints we were able to sequence only three PDXs on and off treatment with cetuximab. To distinguish between the human and the mouse reads a deep sequencing was required, and we run over 40M reads per sample. Importantly, our analysis raised trustful results, as the TGF-beta signature was validated in vivo.  

In Figure 4, the authors claimed that adding CAF led to reduce in the effect of cetuximab “The kinetics of 203 tumor growth (Figure 4A) shows that TGF-beta-activated CAFs limited the efficacy of cetuximab 204 treatment, since cetuximab was less potent in mice with a mix of CAFs and CAL33, compared with 205 tumors in mice with only CAL33.” In fact, this is not clear from the Figure because CAF was able to increase the size of the tumour so it makes sense that the size of the tumour after adding cetuximab is still larger than the group which have cetuximab without CAF. If the claim of the authors is correct, then the authors should measure the percentage of the reduction in the tumour size after adding cetuximab compared with the vehicle group. The same percentage should be calculated in the other group (Call33+CAF-cet VS Call33+CAF-vehcile). Then a comparison between the percentages of the reduction between these two groups should be done.

Answer: We very much appreciate the reviewer's comments regarding the comparison that should be performed between the treatment groups. As the reviewer requested, we performed an additional analysis comparing CAL33 Vehicle and CAL33 Cet versus CAL33 + CAFs Vehicle and CAL33 + CAFs Cet at the endpoint of each treatment group (Figure S4B). Because the untreated groups were very aggressive, mice have sacrificed earlier then the treated groups. Hence, the comparison was done at different time points. Specifically, the presented values in Figure S4B show a normalized tumor volume of untreated groups, thus CAL33 and CAL33-CAFs have an average of 100%. The reduction in tumor volume of the treated groups, were normalized compared to their corresponding untreated group. An unpaired t-test was preform comparing the Cetux-groups with or without CAFs, after normalization to the vehicle groups. The legend for Figure S4 was modified as the following.

Figure S4: In vivo experiments and analysis of CAL33-injected cells with PDX #19 isolated CAFs. (A) The tumor volume of the CAL33 xenograft model with different amounts or without CAFs isolated from PDX #19. 0.5 x 106 cells from each cell line (CAL33 tumor cells, PDX #19 CAFs) were injected subcutaneously in Nod. Scid mice. Mice were randomized into 5 arms (tumors, n = 6-8). The actual tumor volumes ± SEM are presented. Statistical significance was calculated by unpaired t-test (*p < 0.05, **p < 0.01, ***p < 0.001, ****p < 0.0001). (B) Additional analysis of tumor volume of the CAL33 xerograph model with or without CAFs isolated from PDX #19 presented in Figure 4A. Each of the Cetuximab treatment groups was normalized to the respective Vehicle group. Statistical significance was calculated by unpaired t-test (*p < 0.05, **p < 0.01, ***p < 0.001, ****p < 0.0001).

In the method section, the dilution or the working concentration of the antibodies should be mentioned to allow repetition of the work.

Answer: We thank the reviewer for the comment and apologize for missing it from the beginning. We add the antibodies dilution in the materials and method under section 4.7 (Page 13).

There are some grammar mistakes and typos that should be corrected, e.g. “However, such an analysis”, “Forth-four”, and “low ration was”.

Answer: We thank the reviewer for this comment. We sent the paper for a professional English proof editing.

Reviewer 2 Report

Reviewer's report

Title:

TGF-beta-activated cancer-associated fibroblasts limit cetuximab efficacy in preclinical models of head and neck cancer

Version: 1 Date:  2020 Jan 18th

Reviewer's report:

In this manuscript, the Authors investigated the resistance of cetuximab treatment in head and neck cancers using RNA Seq in patient – derived xenografts, identifying overexpression of TGF-beta signaling pathway in cancer associated fibroblasts. Inhibiting SMAD3 by SIS3, a member of TGF-beta pathway, the cetuximab efficacy was enhanced preventing the progression in those mice that showed resistance from this antibody.

.

Data shown are very interesting and deserve to be published. There are only a few minor points to revise:

PTEN loss, KRAS and BRAF mutations must be investigated for all tumors enrolled in this study A Table highlighting the 8 pathways shared by PDX18, PDX03 and PDX20, should be added in paragraph 2.2. Molecular characterization of CetuximabSen and CetuximabProg PDXs; the fold change must be also included. Section Results: Fig.1C: please modify the scale of left graph, which must be the same of the right graph. It looks that there is no difference between the two classes, even if the P was statistically significant. The same for Fig. 3A. Figures 1 and 3 representing the IHC or immunofluorescence in tissues are very small and difficult to observe, even after zooming. They need to be enlarged a little bit without loosing resolution

Round 2

Reviewer 1 Report

The authors addressed most of my comments. Regarding the low sample number for the RNA sequencing, the authors should put this in the limitation of the study as it is an important point in the study design.

Author Response

Dear Reviewer,

Re: Revised version of the manuscript cancers-702471

We thank the reviewer for the comment. We add it as a limitation to our approach in the discussion section as described below.

"In our study, we used an unbiased approach of RNA sequencing to identify the mechanisms of response and resistance to cetuximab. Gene expression profiling of PDXs distinguished between CetuximabSen-PDXs and CetuximabProg-PDX by the changes of stromal genes following cetuximab treatment (Figure 1 and 2). A limitation of our approach is the low number of samples. However, the depth of the RNA-seq allowed the identification of significant candidates that were validated in vitro and in vivo..." (Page 10 in the manuscript).